# CloudformerV3: Multi-Scale Adapter and Multi-Level Large Window Attention for Cloud Detection

**Zheng Zhang** [1,*], **Shuyang Tan** [1] and **Yongsheng Zhou** [2]

1   School of Information, North China University of Technology, Beijing 100144, China; tsy20152090122@mail.ncut.edu.cn
2   College of Information Science and Technology, Beijing University of Chemical Technology, Beijing 100029, China; zhyosh@mail.buct.edu.cn
*   Correspondence: zhangzheng@ncut.edu.cn

**Abstract:** Cloud detection in remote sensing images is a crucial preprocessing step that efficiently identifies and extracts cloud-covered areas within the images, ensuring the precision and reliability of subsequent analyses and applications. Given the diversity of clouds and the intricacies of the surface, distinguishing the boundaries between thin clouds and the underlying surface is a major challenge in cloud detection. To address these challenges, an advanced cloud detection method, CloudformerV3, is presented in this paper. The proposed method employs a multi-scale adapter to incorporate dark and bright channel prior information into the model's backbone, enhancing the model's ability to capture prior information and multi-scale details from remote sensing images. Additionally, multi-level large window attention is utilized, enabling high-resolution feature maps and low-resolution feature maps to mutually focus and subsequently merge during the resolution recovery phase. This facilitates the establishment of connections between different levels of feature maps and offers comprehensive contextual information for the model's decoder. The experimental results on the GF1_WHU dataset illustrate that the method proposed in this paper achieves MIoU of 92.89%, while achieving higher detection accuracy compared to state-of-the-art cloud detection models. Specifically, in comparison to Cloudformer, our method demonstrates a 1.11% improvement, while compared to CloudformerV2, there is a 0.37% increase. Furthermore, enhanced detection performance is achieved along cloud edges and concerning thin clouds, showcasing the efficacy of the proposed method.

**Keywords:** transformer; cloud detection; remote sensing images

## 1. Introduction

Remote sensing technology is assuming an increasingly pivotal role in the realm of Earth observation. However, according to the International Satellite Cloud Climatology Project Flux Data (ISCCP-FD) [1], high-altitude clouds blanket 66% of the Earth's surface, posing a significant impediment to acquiring substantial surface data. This limitation curtails the potential of remote sensing technology. Therefore, within the realm of preprocessing remote sensing images, the detection of clouds assumes paramount importance. By effectively identifying clouds within remote sensing images, surface information can be more accurately extracted. This enhances the authenticity and availability of remote sensing images. Such enhancement holds tremendous practical significance, offering invaluable support for endeavors such as archaeological research [2], the detection of changes in landscape ecology [3], the identification of water bodies [4], weather forecasting [5], and geological landscape analysis [6].

In cloud detection tasks, several challenges persist, such as the difficulty in distinguishing between thin clouds and the Earth's surface, the tendency for bright surfaces like ice and snow to be mistakenly identified as cloud layers, and the challenges in detecting small and fragmented clouds. Therefore, addressing the issue of unclear boundaries between

thin clouds and the Earth's surface in complex scenarios, this paper continues to delve into the application of the transformer in the cloud detection task. Based on Cloudformer [7] and CloudformerV2 [8], this study suggests using CloudformerV3 to recognize clouds in high-resolution remote sensing image data.

The three important areas covered by CloudformerV3's principal contributions are as follows:

1.  Multi-Scale Adapter Incorporation in the Encoder: Introducing a multi-scale adapter within the encoder enhances the synergy between the pretrained natural image-based backbone and the remote sensing image cloud detection process. This collaboration facilitates multi-level feature extraction, allowing the model to gain a more profound understanding of image structure and characteristics that are pertinent to the task. Furthermore, through the use of an adapter, downstream tasks and other prior information can be integrated into the model. This allows for the adaptation of multi-channel data information without compromising the use of models pretrained on a substantial amount of natural images;

2.  Multi-Level Large Window Attention Enhanced Decoder Mechanism: In the decoder stage, a novel approach is adopted involving the interaction of low-resolution feature maps with high-resolution feature maps through large window attention, which establishes connections across different levels of feature maps, thereby enabling the model to enhance contextual understanding. This process encompasses incremental layer-by-layer upsampling and fusion with higher-level feature maps. As a result, the decoder comprehensively integrates feature information across multiple levels, thereby intensifying the ability to detect cloud edges;

3.  Integration of Dark and Bright Channel Prior to Information: During the data preprocessing phase, the dark channel and bright channel prior information are computed and then integrated into the generalized backbone through an adapter. This infusion equips the model with prior feature information that significantly enhances cloud detection capabilities. In particular, this enhancement contributes to distinguishing thin clouds from the ground surface with greater precision.

The remainder of this paper is organized as follows. Section 2 introduces the related work. Section 3 presents the design details of our proposed network. Section 4 provides the relevant experiments and setups. Section 5 summarizes our approach, and Section 6 presents the outlook.

## 2. Related Work

The traditional cloud detection method is based on spectral thresholds, which utilize the unique physical characteristics of clouds to construct multiple spectral thresholds. Among them, Automated Cloud Cover Assessment [9,10] (ACCA) and Function of Mask (Fmask) [11] are the most representative methods. However, methods based on spectral thresholds typically require remote sensing images to possess rich spectral information. Therefore, these methods are often applied to imagery from the Landsat series or Sentinel-2 satellites, which provide extensive spectral data. For remote sensing images with a limited number of spectral bands or those consisting primarily of visible light, such algorithms may face challenges in effective operation. Moreover, methods based on spectral thresholds often exhibit poor generalization capabilities, with limited robustness, particularly in diverse and complex scenes [12,13]. Modern cloud detection technology is usually based on classic machine learning methods, using algorithms such as Support Vector Machine (SVM) [14] and Random Forest (RF) [15] to identify the cloud. The prediction time complexity for SVM is generally $O(d)$, where $d$ is the number of features. And the prediction time complexity for each tree is typically $O(log(n))$, where $n$ is the number of nodes in the tree. These methods make better use of spatial information from remotely sensed images and also reduce the high dependence of remotely sensed image data on spectral range. However, due to the fact that classical machine learning models typically require manual design of image features,

this method is difficult to effectively extract higher-level semantic information and has poor robustness to complex scenes [13].

In recent years, deep neural networks have shown good performance in segmentation tasks due to their powerful feature extraction capabilities. This method has also been applied to remote sensing image cloud detection and achieved good results. Inspired by convolutional neural networks for semantic segmentation such as FCN [16], SegNet [17], UNet [18], and DeepLabV3+ [19], Francis et al. proposed CloudFCN [20], a method of cloud detection in combination with Inception module, while Jeppesen et al. proposed RSNet (Remote Sensing Network) [21] for remote sensing images in RGB. As researchers delve deeper into the realm of cloud detection tasks, they have recognized that devising effective methods tailored to the distinct characteristics and challenges of cloud detection tasks stands as a pivotal technology for enhancing algorithm precision. Yang and colleagues introduced CDNet [22], a solution designed for low-resolution remote sensing thumbnail images. This innovation bolsters the accuracy of detecting clouds in low-resolution images through edge refinement and the integration of a feature pyramid structure. Li et al. focused their efforts on high-resolution remote sensing images and devised MSCFF [23], which employs a multi-scale feature fusion approach. Guo and his team proposed CDNetV2 [24], a model that attains high-precision cloud detection, even in scenarios where clouds and snow coexist. In 2022, He et al. proposed a lightweight network, DABNet [25], which creates not only a lower false-alarm rate but also a clearer boundary. In the same year, to effectively detect thin clouds, Li et al. proposed a novel robust cloud detection approach, GCDB-UNet [26], which embeds GCDB (global context dense block) to UNet. The time complexity of convolutional neural network models generally depends on the structure of the model. When the number of input pixels is $n$, the time complexity of one convolutional operation is usually $O(n \times K^2 \times C)$, where $K$ is the size of the convolutional kernel and $C$ is the number of output channels. Typically, convolutional neural networks have higher time complexity compared to traditional models.

Since 2018, the transformer [27] has achieved great success in the field of natural language processing and has gradually been applied to image segmentation. The transformer has powerful feature extraction capabilities, which can simultaneously model both global and local features of an image. These abilities enable it to perform well in semantic segmentation, and many high-precision semantic segmentation models have emerged, such as SETR [28], SegFormer [29], MaskFormer [30], Mask2Former [31], Lawin Transformer [32], ViT Adapter [33], and Mask DINO [34]. The time complexity of the transformer model mainly considers the self-attention and feedforward neural networks. When the number of input pixels is $n$, the actual complexity of the self-attention is usually $O(n \times L)$, where $L$ is the length of the input sequence. For feedforward neural networks, the time complexity is usually proportional to the size of the input and the number of weight parameters. Transformers have higher time complexity compared to convolutional neural networks in general. However, when used for segmentation tasks, these models are usually easier to achieve higher accuracy than convolutional neural network models. When directly applying the aforementioned models to the task of cloud detection in remote sensing images, certain challenges persist. These include difficulties in distinguishing between thin clouds and surfaces, the intricacy of detecting minute cloud formations, and the potential for misidentifying bright surfaces or icebergs as clouds. To address these issues, it becomes imperative to enhance or reconfigure models tailored to natural image data, enabling their effective application to remote sensing image cloud detection tasks. A cloud detection method CloudViT [35] for a lightweight vision transformer was proposed by Zhang et al. in 2023, which improves the vision transformer and improves model accuracy while maintaining a small computational and parameter size. Our previous iterations such as Cloudformer [7] and CloudformerV2 [8] have effectively tackled the challenge of detecting small clouds, resulting in improved cloud detection accuracy and faster training inference.

The overarching framework of related work is shown in Figure 1.

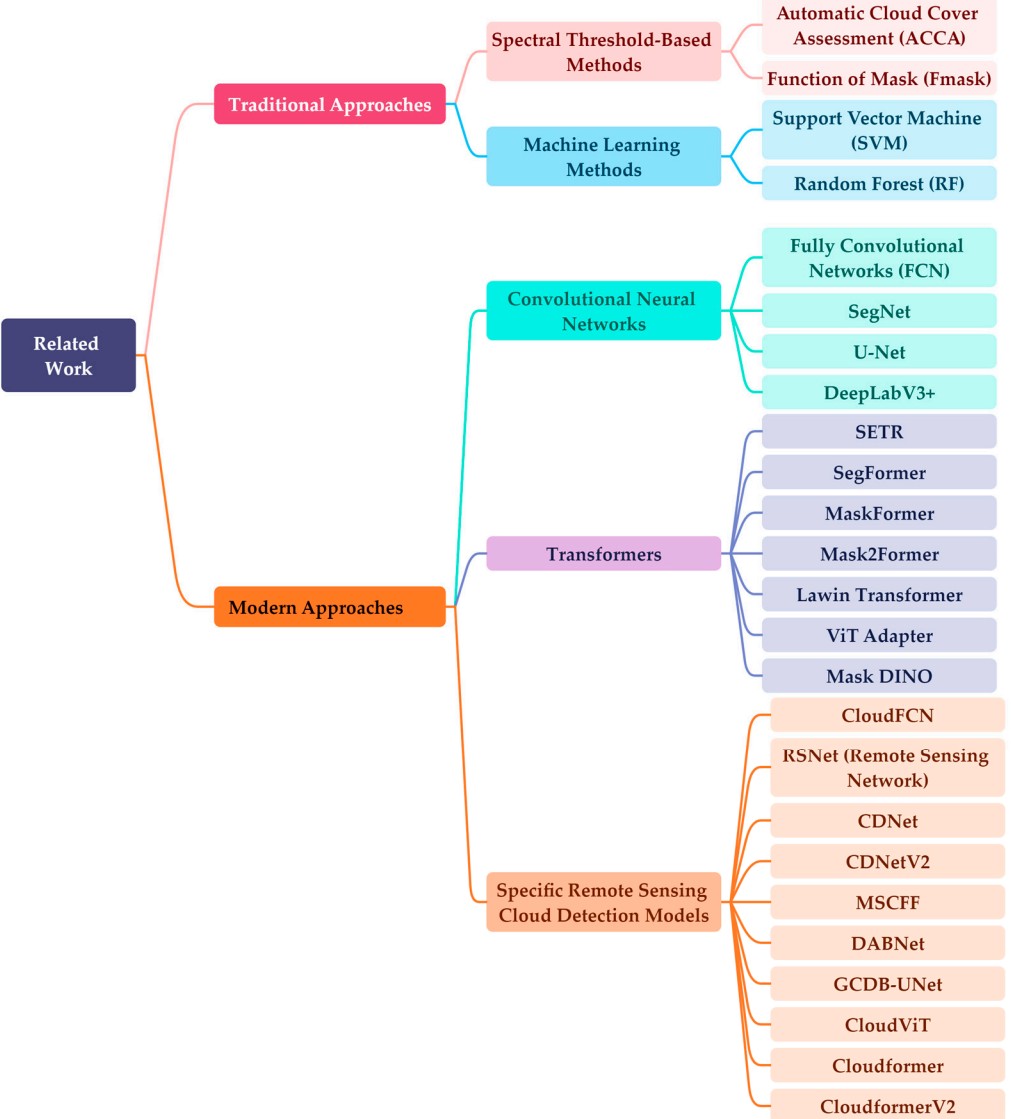

**Figure 1.** The overarching framework of related work, delineated into two categories: traditional approaches and modern approaches. Traditional approaches comprise spectral threshold-based methods and machine learning methods. On the other hand, modern approaches encompass convolutional neural networks, transformers, and specific remote sensing cloud detection models.

## 3. Materials and Methods

### 3.1. Overall Structure of CloudformerV3

The structure of CloudformerV3 is composed of two main components: the encoder and the decoder, as depicted in Figure 2. Within the encoder, the backbone incorporates the Mix Transformer [29], optimizing the extraction of image features and striking a balance between accuracy and efficiency. During the fine-tuning process with cloud datasets, a multi-scale adapter is introduced to work in conjunction with the backbone for downsampling operations. Concurrently, the prior information from both dark and bright channels is combined and infused into the backbone through the adapter, fostering comprehensive interaction between the prior information and the input image. The encoder obtains the multi-level feature layer and channels it to the decoder. This enables distinct feature layers at varying levels to mutually focus on one another through a multi-level large window attention. Subsequently, during the resolution recovery phase, the decoder progressively upsamples and integrates the multi-level feature layer with higher-level feature layers. This culminates in the generation of the segmentation result.

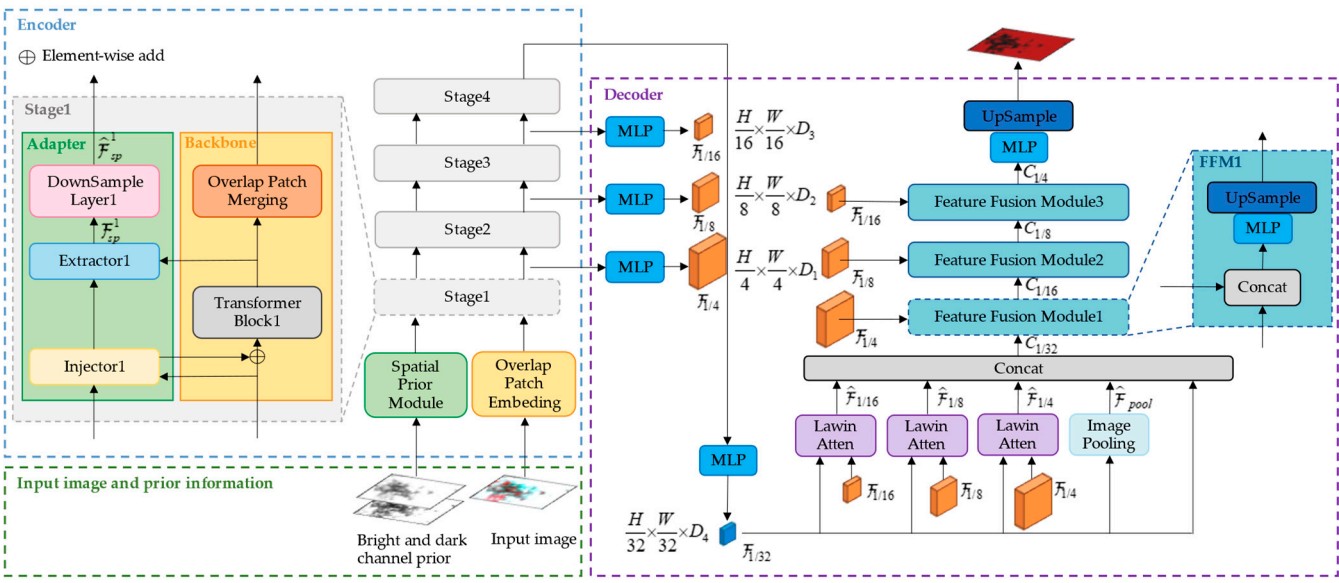

**Figure 2.** The overall structure of CloudformerV3 consists of three main components: Green dashed outline represents the input image and prior information, bule dashed outline represents the encoder and purple dashed outline represents the decoder.

### 3.2. Multi-Scale Adapter

The traditional adapter [33] comprises three components: Spatial Prior Module, Injector, and Extractor. Spatial Prior Module is used to extract spatial information from the image, while Injector and Extractor modules facilitate interaction with the backbone. When applied to cloud detection, due to the substantial disparities between natural images and remote sensing images, using the backbone trained on natural images directly for fine-tuning yields unsatisfactory outcomes. In contrast, the adapter can infuse prior information from remote sensing images into the backbone trained on natural images, rendering the model better suited for cloud detection. This adaptation enhances the model's effectiveness in cloud detection tasks.

However, the traditional adapter cannot be applied to the hierarchical backbone structure, which makes the multi-scale feature extraction suffer. To solve this problem, this paper proposes a novel multi-scale adapter that inserts DownSample Layer between each Extractor and Injector, as shown in Figure 3. After the $i$th Extractor, it can obtain the output feature $F_{sp}^i \in \mathbb{R}^{(HW/(2i)^2 + HW/(4i)^2 + HW/(8i)^2) \times D_i}$. In the DownSample Layer, it is first split and reshaped to obtain three feature maps $F_{sp1}^i \in \mathbb{R}^{H/(2i) \times W/(2i) \times D_i}$, with different resolutions. Then the feature map is reduced to one-half of its original length and width by the Patch Merging layer [36] to obtain $\hat{F}_{sp1}^i \in \mathbb{R}^{H/(4i) \times W/(4i) \times D_{i+1}}$, $\hat{F}_{sp2}^i \in \mathbb{R}^{H/(8i) \times W/(8i) \times D_{i+1}}$, and $\hat{F}_{sp3}^i \in \mathbb{R}^{H/(16i) \times W/(16i) \times D_{i+1}}$. Finally, $\hat{F}_{sp1}^i$, $\hat{F}_{sp2}^i$, and $\hat{F}_{sp3}^i$ are flattened and concatenated to obtain $\hat{F}_{sp}^i \in \mathbb{R}^{(HW/(4i)^2 + HW/(8i)^2 + HW/(16i)^2) \times D_{i+1}}$, which is injected into the next Injector. The process can be formulated as:

$$F_{sp1}^i, F_{sp2}^i, F_{sp3}^i = Reshape\left(Split(F_{sp}^i)\right) \tag{1}$$

$$\hat{F}_{spj}^i = PatchMerging(F_{spj}), j \in \{1, 2, 3\} \tag{2}$$

$$\hat{F}_{sp}^i = Flatten\left(Concat(\hat{F}_{sp1}^i, \hat{F}_{sp2}^i, \hat{F}_{sp3}^i)\right) \tag{3}$$

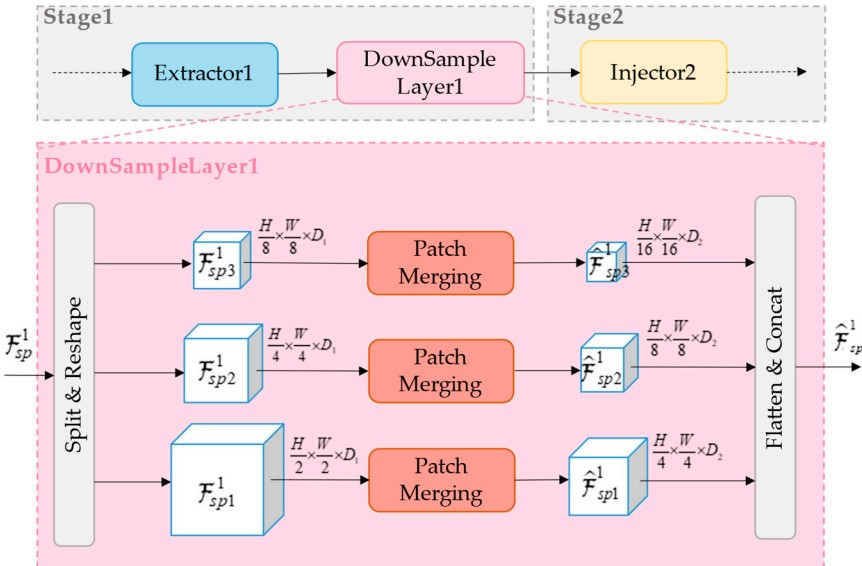

**Figure 3.** The structure of DownSample Layer. Bule represents the extractor, pink represents the downsample layer and yellow represents the injector.

The utilization of a multi-scale adapter accomplishes more than just enhancing the model's suitability for the cloud detection task and introducing prior information. It also empowers the backbone to acquire precise prior information across all scales, enabling a more comprehensive grasp of the image's structure and features. This facilitates improved differentiation between regions containing clouds and those without, particularly in intricate scenarios.

### 3.3. Multi-Level Large Window Attention and Decoder

Detecting thin clouds in remote sensing images is frequently intricately linked to the surrounding context. The employment of multi-level feature maps extends a broader array of contextual insights, equipping the model with an enhanced capacity to comprehend the intricate interplay between thin clouds and their neighboring features. Consequently, this advanced comprehension facilitates a more precise identification of thin clouds. Based on the above reasons, this paper designs a novel decoder in conjunction with the multi-level large window attention, so that the high-level feature maps and the low-level feature maps can pay attention to each other and merge with each other.

Inspired by the concept of large window attention proposed in the Lawin Transformer [32], this paper introduces a multi-level large window attention. In the context of the large window attention, a feature graph is divided into uniformly sized windows, allowing each window denoted as Q to query a larger region represented by C. However, within the framework of the multi-level large window attention, the low-resolution feature map is segmented into $n \times n$ windows to replace the query window Q, while the high-resolution feature map undergoes a similar partitioning into $n \times n$ windows to substitute the queried region C. This transforms the operation of focusing on a single feature map into the interaction between feature maps at different levels, as illustrated in Figure 4. As the window slides across the low-resolution feature map, the current segment is permitted to query the corresponding region within the high-resolution map, encompassing more pixel information. This capability enables the decoder to establish connections between feature maps at varying levels, thereby furnishing the model with a more extensive and intricate contextual understanding.

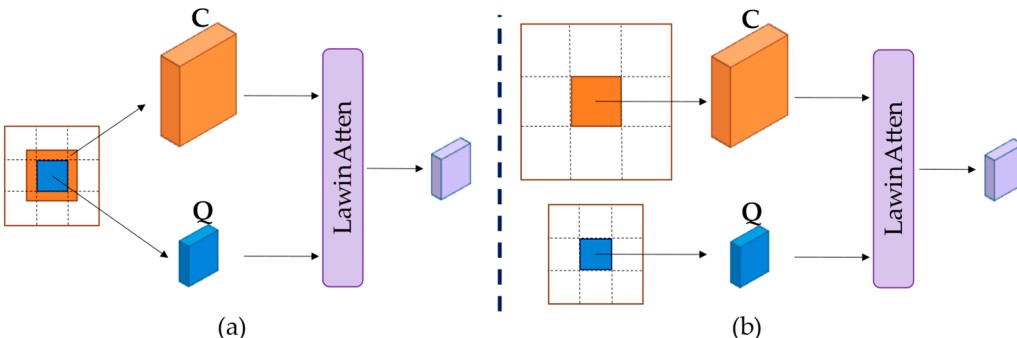

**Figure 4.** Comparison of Large Window Attention and Multi-level Large Window Attention. (**a**) Large Window Attention. (**b**) Multi-Level Large Window Attention.

In the decoder, the multi-level feature maps of 1/4, 1/8, 1/16, and 1/32 of the input image size in length and width obtained after passing through the encoder are passed through the multilayer perceptron, respectively, to obtain $F_{1/4} \in \mathbb{R}^{H/4 \times W/4 \times D_1}$, $F_{1/8} \in \mathbb{R}^{H/8 \times W/8 \times D_2}$, $F_{1/16} \in \mathbb{R}^{H/16 \times W/16 \times D_3}$, and $F_{1/32} \in \mathbb{R}^{H/32 \times W/32 \times D_4}$. In this paper, five parallel branches are designed in conjunction with multi-level large window attention, including a skip connection, a branch pooling $F_{1/32}$ to obtain $\hat{F}_{pool} \in \mathbb{R}^{H/32 \times W/32 \times D_4}$, and the three branches will be $F_{1/16}$, $F_{1/8}$, and $F_{1/4}$, respectively, with $F_{1/32}$ in multi-level large window attention to obtain $\hat{F}_{1/4} \in \mathbb{R}^{H/32 \times W/32 \times D_4}$, $\hat{F}_{1/8} \in \mathbb{R}^{H/32 \times W/32 \times D_4}$, and $\hat{F}_{1/16} \in \mathbb{R}^{H/32 \times W/32 \times D_4}$. All the obtained features are stitched together to obtain a feature map with multi-level information. The process can be formulated as:

$$\hat{F}_{pool} = Pooling(F_{1/32}) \tag{4}$$

$$\hat{F}_i = LawinAtten(F_i, F_{1/32}), i \in \{1/4, 1/8, 1/16\} \tag{5}$$

$$C_{1/32} = Concat(\hat{F}_{1/4}, \hat{F}_{1/8}, \hat{F}_{1/16}, \hat{F}_{pool}, F_{1/32}) \tag{6}$$

$C_{1/32}$ is upsampled layer by layer through the three feature fusion modules and in the process fused with the high-level feature maps $F_{1/16}$, $F_{1/8}$, and $F_{1/4}$, respectively, spliced to recover the image resolution to obtain $C_{1/4}$. Finally, the result is output after passing through the multilayer perceptron and upsample layer.

### 3.4. Dark and Bright Channel Prior Information

Because of the visual resemblance between thin clouds and haze, cloud detection tasks exhibit similarities with tasks involving the dehazing of remote sensing images [37]. In the realm of image dehazing, the dark channel prior [38] and the bright channel prior [39] hold substantial importance. During the preprocessing phase of this research, alongside utilizing image information from the three RGB channels, both the dark channel and the bright channel priors are integrated to aid the detection model. For a given image, the dark channel is defined as:

$$I^{dark}(x) = \min_{c \in \{r,g,b\}} \left( \min_{y \in \Omega(x)} (I^c(y)) \right) \tag{7}$$

while the bright channel is defined as:

$$I^{bright}(x) = \max_{c \in \{r,g,b\}} \left( \max_{y \in \Omega(x)} (I^c(y)) \right) \tag{8}$$

where $I^c$ represents the RGB color mode of the image $I$, $\Omega(x)$ denotes a local region centered at $x$, and $y$ represents a pixel point in the image $I$. When extracting the dark channel and

the bright channel, to retain as much original information as possible, the size of $\Omega(x)$ is chosen as $1 \times 1$. This is because image segmentation is an end-to-end task.

Figure 5 provides an illustration of the dark channel and the bright channel for the same scene. In remote sensing images containing cloud layers, both thin and thick clouds have pixel points $x$ with values of $I^{bright}(x) \rightarrow 255$, resulting in a more complete image structure at the edges of clouds in the bright channel compared to the dark channel. In the dark channel, the value of $I^{dark}(x)$ in cloud-obscured regions is notably distinct from that in non-cloud regions, indicating that the dark channel enhances the brightness contrast between the cloud-obscured and non-cloud areas, facilitating a clearer differentiation between clouds and background information. Therefore, this study concatenates the dark channel and the bright channel for input into the model, allowing them to complement each other's strengths and weaknesses, thereby providing a more comprehensive and enriched set of image information.

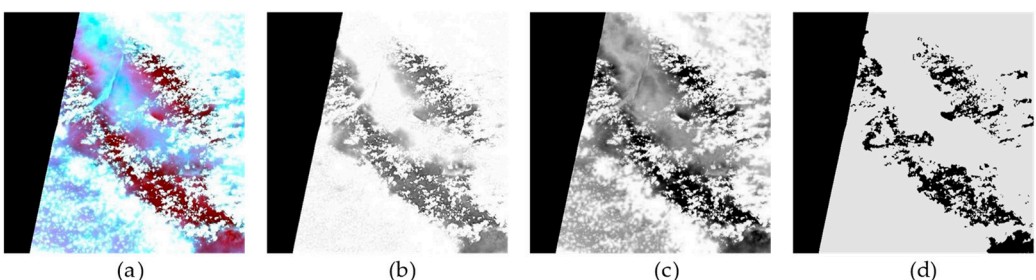

(a)  (b)  (c)  (d)

**Figure 5.** An example of the bright channel and dark channel. (**a**) Input image. (**b**) Bright channel. (**c**) Dark channel. (**d**) Ground Truth.

## 4. Results

### 4.1. Dataset

High-resolution remote sensing image data possess characteristics such as high spatial resolution and detailed terrain features, often making them sensitive to cloud and fog presence. Hence, for the experimental dataset, this study selected the GF1_WHU dataset [40]. Acquired from May 2013 to August 2016, the dataset comprises 108 scenes of GF-1 Wide Field of View (WFV) 2A-level images along with their corresponding reference masks. These reference masks are manually drawn through visual inspection, resulting in more precise delineation of cloud and cloud shadow boundaries, which facilitates better model fitting. Example graph of the dataset is shown in Figure 6. To reduce computational complexity, this study employed RGB channel image thumbnails. Given the focus on cloud detection, the dataset was further processed to remove cloud shadow annotations. Ultimately, it was resized to 777 images of dimensions 512×512 for both training and testing. Among them, 538 images were allocated for training, while an additional 239 images were set aside for testing.

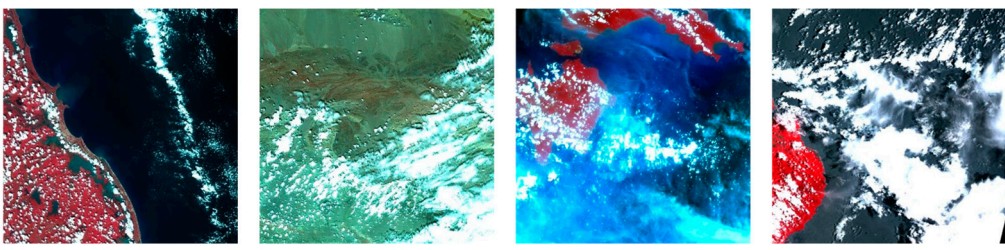

**Figure 6.** This figure illustrates representative instances from the dataset, showcasing images with varying cloud cover across different scenes.

*4.2. Experimental Environment*

The experimental computations were conducted using the powerful Nvidia GeForce RTX 2080 Ti graphics processing unit (GPU). The GPU was integrated into a system running Linux, and the experiments were conducted using PyTorch 1.8.1 framework.

*4.3. Training and Testing Process*

We employed the cross-entropy loss function and the AdamW optimizer, coupled with a polynomial learning rate adjustment strategy and linear warm-up at the beginning, which aids in effectively smoothing the convergence and stabilizing the model during the initial training phases. Throughout the training process, we closely monitored the loss function values to assess the model's convergence. When the model's performance became stable, and the loss values exhibited no significant fluctuations, training was concluded. In the testing stage, we conducted a thorough evaluation using comprehensive metrics, including mean intersection over union (*MIoU*) [41], mean accuracy (*MAcc*) [42], and pixel accuracy (*PAcc*) [43], to assess the model's overall performance on the test dataset. These metrics offered profound insights into the model's applicability and effectiveness. These formulas define the evaluation metrics:

$$MIoU = \frac{1}{k+1}\sum_{i=0}^{k}\frac{TP}{FN+FP+TP} \qquad (9)$$

$$MAcc = \frac{1}{k+1}\sum_{i=0}^{k}\frac{TP}{FP+TP} \qquad (10)$$

$$PAcc = \frac{TP+TN}{TP+TN+FP+TN} \qquad (11)$$

where *k* represents the number of categories. Due to cloud detection being a binary classification problem, *k* = 1 during the computation process. *TP* (True Positive) signifies the count of pixels correctly predicted as clouds by the model, *TN* (True Negative) represents the count of pixels correctly predicted as background, *FP* (False Positive) stands for the count of misclassified pixels predicted as clouds, and *FN* (False Negative) denotes the count of missed pixels predicted as background.

*4.4. Ablation Experiment*

4.4.1. Performance Verification of the Multi-Scale Adapter

To explore the impact of the multi-scale adapter for the model, a quantitative analysis was conducted by comparing the performance of models without the adapter and those with the multi-scale adapter. And to avoid the influence of prior information on dark and bright channels, the input image information only includes RGB channels. The results are presented in Table 1. It can be observed that the model with the introduced multi-scale adapter exhibits stronger adaptability in terms of network structure. This enables better integration of prior information from remote sensing images, resulting in superior performance in cloud detection tasks. Consequently, the model's accuracy in cloud detection is effectively enhanced.

**Table 1.** Performance comparison of models before and after adding the multi-scale adapter.

| Encoder | MIoU (%) | MAcc (%) | PAcc (%) |
|---|---|---|---|
| Mix transformer | 91.63 | 94.97 | 96.44 |
| **+Multi-Scale Adapter** | **92.43** | **95.62** | **96.95** |

Furthermore, a comparative analysis of images processed by models without the adapter and models with the multi-scale adapter was conducted, as shown in Figure 7. Through this analysis, it becomes evident that images processed with the adapter exhibit

notably improved detection performance at cloud edges. The differentiation between thin clouds and the background aligns more closely with the true label values. This observation suggests that the model's perceptual capabilities are enhanced, consequently improving its accuracy and robustness in practical applications. This further validates the effectiveness of the multi-scale adapter in cloud detection tasks.

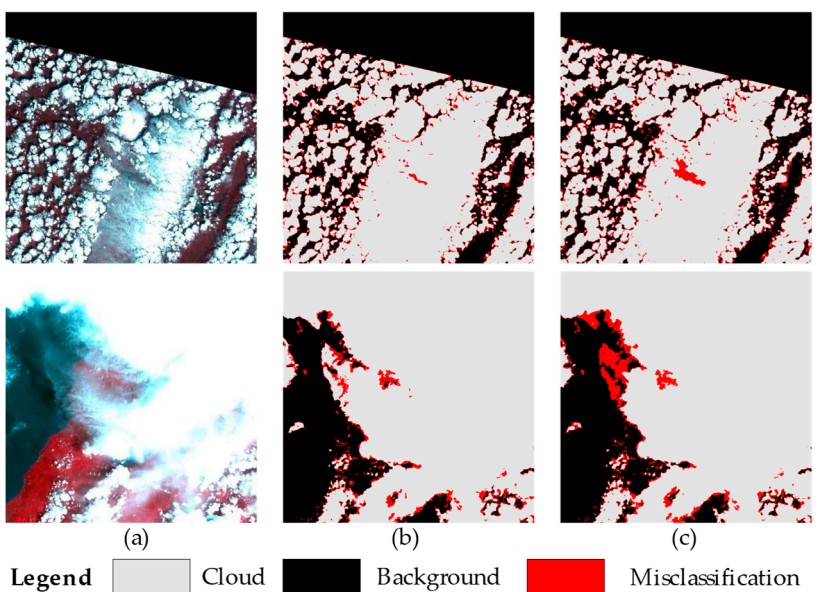

| | | | Cloud | | Background | | Misclassification |
|---|---|---|---|---|---|---|---|
| (a) | | (b) | | | (c) | | |

**Legend**  ▢ Cloud  ■ Background  ■ Misclassification

**Figure 7.** Comparison of models before and after adding multi-scale adapter. (**a**) Input image. (**b**) Multi-Scale Adapter. (**c**) Baseline.

### 4.4.2. Performance Validation of Multi-Level Large Window Attention

To validate the performance of the decoder using the multi-level large window attention, this study replaced the large window attention for both low-resolution feature layers and high-resolution feature layers with attention solely driven by high-resolution feature layers. Furthermore, in order to ensure consistent feature map sizes, bilinear interpolation was employed to downsample the high-resolution feature layers used by the latter approach. The results obtained from the experiment are presented in Table 2. These results indicate that using the multi-level large window attention with integration of both low-level semantic information and high-level semantic information yields higher model accuracy.

**Table 2.** Performance comparison between single-level layer large window attention and multi-level large window attention.

| Decoder | MIoU (%) | MAcc (%) | PAcc (%) |
|---|---|---|---|
| Single feature layer | 92.26 | 95.54 | 96.83 |
| **Multi-feature layer** | **92.89** | **96.04** | **97.12** |

The segmentation results of the two models were also compared in this study, as illustrated in Figure 8. It can be observed that, compared to employing the large window attention on a single feature map, conducting the large window attention operation between low-level and high-level feature maps leads to a significant enhancement in detecting cloud edges. This observation highlights the effectiveness of the multi-feature layer large window attention in considering information from different levels, resulting in improved cloud detection performance by the model.

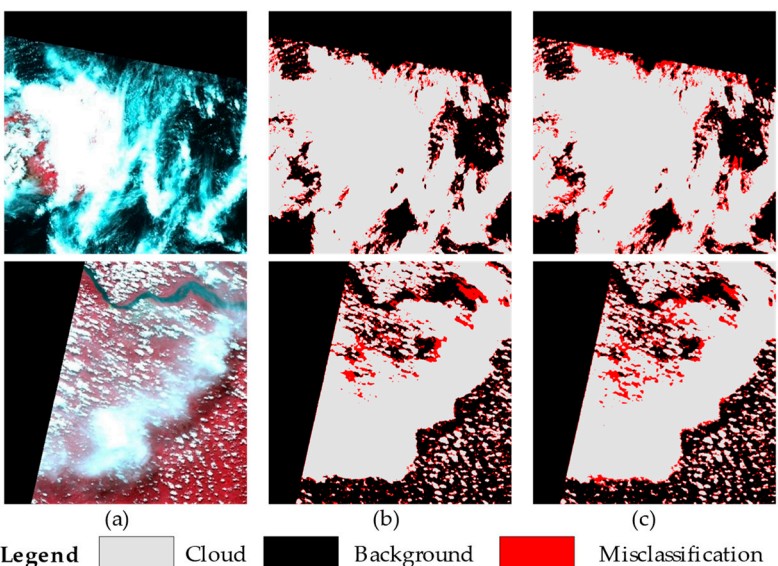

**Legend** Cloud ■ Background ■ Misclassification

**Figure 8.** Comparison of models with single feature layer and multi-feature layer large window attention. (**a**) Input image. (**b**) Multi-feature Layer. (**c**) Single feature Layer.

4.4.3. Performance Validation of Dark Channel and Bright Channel Prior information

To validate the effectiveness of the dark channel and bright channel prior information in remote sensing image cloud detection, this study introduced various combinations of different channels into the input of the adapter and conducted corresponding experiments and analyses. While retaining the RGB channel image as the input to the backbone, seven different image data input methods were introduced in the adapter: RGB channels, dark channel, bright channel, concatenation of dark channel and RGB channels, concatenation of bright channel and RGB channels, concatenation of dark channel and bright channel, and concatenation of dark channel, bright channel, and RGB channels. The experimental results are presented in Table 3.

**Table 3.** Comparison of effects from different channels input image data.

| Input Channel | MIoU (%) | MAcc (%) | PAcc (%) |
|---|---|---|---|
| RGB | 92.43 | 95.62 | 96.95 |
| Dark | 92.71 | 95.77 | 97.09 |
| Bright | 92.68 | 95.83 | 97.05 |
| Dark + RGB | 92.33 | 95.48 | 96.91 |
| Bright + RGB | 92.14 | 95.51 | 96.84 |
| **Dark + Bright** | **92.89** | **96.04** | **97.12** |
| Dark + Bright + RGB | 92.03 | 95.45 | 96.78 |

From the experimental results, the following trends can be observed: When combining the dark channel and bright channel prior information and inputting them into the adapter, the model achieves the highest detection accuracy. This indicates that the fusion of these two types of prior information in remote sensing image cloud detection tasks provides the model with more comprehensive and accurate information, leading to superior performance. Furthermore, the experimental results for separately inputting the dark channel or bright channel prior information follow, with both demonstrating better performance compared to solely inputting RGB channels. This result emphasizes the role of the dark channel and bright channel prior information in enhancing model performance. However, a decline in accuracy is observed when attempting to concatenate the RGB channels with the prior information and inputting them into the adapter. This could be attributed to conflicting information or increased model complexity, leading to poorer interaction between channels, consequently affecting the effectiveness of the prior information.

*4.5. Comparison with State-of-the-Art Methods*

The experiments conducted on the GF1_WHU dataset reveal that CloudformerV3 surpasses existing advanced methods in terms of accuracy. Meanwhile, we compared the speed at which different models can infer an image with a resolution of 512 × 512, as illustrated in Table 4. This underscores CloudformerV3's remarkable advantage in terms of segmentation precision. Furthermore, in real-world detection scenarios, CloudformerV3 demonstrates significant enhancement in segmentation performance, as depicted in Figure 9.

**Table 4.** Comparison of performance with state-of-the-art methods. Indicators with ↑ prefer higher values, while those with ↓ favor lower values.

| Method | Evaluation Metrics | | | Inference Time (ms) ↓ |
|---|---|---|---|---|
| | MIoU (%) ↑ | MAcc (%) ↑ | PAcc (%) ↑ | |
| SwinTransformer-UperNet | 90.47 | 93.37 | 94.12 | 12.07 |
| Mask2former | 90.89 | 94.69 | 94.89 | 13.84 |
| Segformer | 90.65 | 94.92 | 95.61 | 8.40 |
| Lawin transformer | 90.73 | 94.55 | 95.67 | 10.92 |
| GCDB-UNet | 89.45 | 93.62 | 94.08 | 7.83 |
| ViT Adapter-UperNet | 91.85 | 95.20 | 96.11 | 18.87 |
| Cloudformer | 91.78 | 94.49 | 95.07 | 11.20 |
| CloudformerV2 | 92.52 | 95.66 | 96.75 | 14.29 |
| **CloudformerV3** | **92.89** | **96.04** | **97.12** | **15.54** |

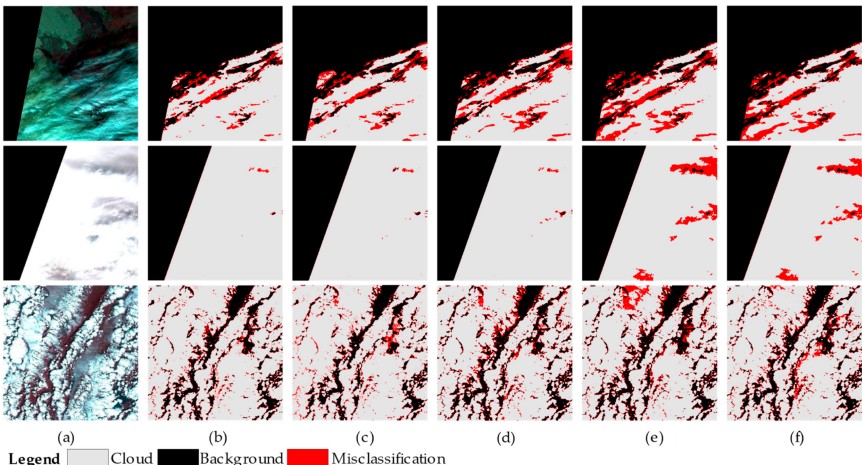

**Legend** ☐ Cloud ■ Background ■ Misclassification

**Figure 9.** Versus different methods using the GF1_WHU dataset. (**a**) Image. (**b**) CloudformerV3. (**c**) CloudformerV2. (**d**) Cloudformer. (**e**) Lawin transformer. (**f**) Segformer.

This model adeptly captures cloud textures, shapes, and contextual information, leading to more precise and refined segmentation of delicate cloud segments. These outcomes underscore the exceptional performance of CloudformerV3 in cloud detection within high-resolution remote sensing images, offering a dependable solution for cloud detection tasks within the domain of remote sensing imagery applications.

**5. Conclusions**

This paper addresses the challenge of distinguishing between thin clouds and the unclear boundaries of surfaces in complex scenes, proposing an effective cloud detection method named CloudformerV3 tailored for high-resolution remote sensing image data. By introducing a multi-scale adapter into the backbone, this method accomplishes dual objectives: injecting prior information from remote sensing images into the backbone to enhance its suitability for remote sensing image cloud detection tasks, and enabling multi-scale

feature extraction in collaboration with the backbone to capture richer image information. Additionally, the decoder employs multi-level large window attention, facilitating the establishment of connections between feature maps from different levels. In the data preprocessing stage, dark channel and bright channel prior information is incorporated, enabling the model to acquire more comprehensive prior knowledge. This approach offers a strategy to integrate multi-channel remote sensing image prior information into a model pretrained on natural images. Compared to existing advanced models, CloudformerV3 exhibits superior performance on the GF1_WHU dataset, boasting higher accuracy and effectively enhancing its ability to detect thin clouds and cloud edges. These findings validate for effective cloud detection of CloudformerV3 in optical remote sensing image.

### 6. Future Perspectives

While CloudformerV3 has showcased impressive performance on the GF1_WHU dataset, it is imperative to acknowledge its inherent limitations. The integration of sophisticated elements like the multi-scale adapter and multi-level large window attention introduces potential challenges in terms of computational expenses. Additionally, despite its outstanding performance on high-resolution imagery, there is a need for further validation regarding its generalization capabilities across diverse datasets and varied geographical settings.

Potential Directions for Future Research:

1. Enhancing Computational Efficiency: In order to mitigate computational costs, future research endeavors can focus on refining the computational efficiency of CloudformerV3. This may involve exploring more lightweight network architectures or optimizing algorithms;
2. Investigating Generalization Performance: Further studies should delve into the generalization performance of CloudformerV3 across different datasets and geographical environments, ensuring its robustness and reliability in practical applications;
3. Integration of Multi-Source Data: Considering the integration of diverse remote sensing data sources, such as infrared or radar data, into CloudformerV3 could enhance its adaptability to multi-modal data, broadening its applicability.

**Author Contributions:** Conceptualization, Z.Z. and S.T.; methodology, Z.Z. and S.T.; software, S.T.; validation, Z.Z. and S.T.; formal analysis, S.T. and Y.Z.; investigation, Z.Z.; resources, Z.Z. and Y.Z.; data curation, Z.Z. and S.T.; visualization, S.T.; writing original draft preparation, S.T.; writing review and editing, Z.Z. and Y.Z.; supervision, Z.Z.; project administration, Z.Z. All authors have read and agreed to the published version of the manuscript.

**Funding:** This research received no external funding.

**Institutional Review Board Statement:** Not applicable.

**Informed Consent Statement:** Not applicable.

**Data Availability Statement:** Data are contained within the article.

**Conflicts of Interest:** The authors declare no conflict of interest.

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
