# Peer review of "CloudformerV3: Multi-Scale Adapter and Multi-Level Large Window Attention for Cloud Detection"

_applsci, doi:10.3390/app132312857_

Round 1

Reviewer 1 Report

Comments and Suggestions for Authors

This manuscript presents an automatic cloud detection methods from satellite images. Overall, the topic is closely related to the journal and the manuscript is well structured. The description of the presented methodology is clear as well. There are only a few minor suggestions and comments to the manuscript of current form.

1. It is suggested to add a brief quantitative description of the validation results in abstract.

2.  Captions of the figures should be self-explanatory. It is recommended to add detailed descriptions to the captions of the figures throughout the manuscript.

3.  There are some minor typos and grammar errors, please check the entire text.

Comments on the Quality of English Language

There are only a few minor typos and errors in English grammar.

Reviewer 2 Report

Comments and Suggestions for Authors

The topic is interesting and relevant but needs to answer some questions and some modifications are

1. The numerical achievements of the research should be brought at the end of the abstract section.

2.   In the introduction section, the authors stated that methods based on spectral thresholds often have poor generalization ability and overly rely on spectral information. Adding a reference that confirms this claim is suggested.

3. The novelty of the research should be explained in more detail.

4. The research's main contributions should be highlighted at the end of the introduction and before the organization of the paper.

5. Authors are suggested to compare their proposed model with some new and state-of-the-art algorithms.

6. The time complexity and order of the algorithms should be added to the introduction section.

7. It is suggested to separate the literature review from the introduction section.

8. The overhead and limitations of the proposed method should be discussed.

9. The speed of the proposed method should be compared with other methods.

10. The simulation environment should be determined.

11. The algorithm parameters and the parameter tuning methods have not been considered. 

12. Adding direction for potential future suggestions is suggested.

13. Adding more recent papers( 2022 and 2023) to the literature review section is suggested.

Reviewer 3 Report

Comments and Suggestions for Authors

The paper describes a cloud detection method named CloudformerV3 tailored for high-resolution remote sensing image data and shows that compared to existing advanced models, CloudformerV3 exhibits superior performance on the GF1_WHU dataset. 

The introduction is very informative and provides a lot of needed information, although the reference to Fig. 1 should be inserted into the text.

The new method is well described, although it is not clear how to understand the training and testing process.

Lines 245-246: It is said, 777 images were used for training and testing. However, there is no information on how they were divided for training and testing sets.

Some small editorial flaws:

line 111: "be-tween" should be changed to "between"

line: 178: "n × n" - the font type can be corrected to make it consistent with the one presented in the next line of text

There is no reference to Fig.6 (as well as Fig.1, as mentioned earlier)

Round 2

Reviewer 2 Report

Comments and Suggestions for Authors

The authors addressed all my previous concerns in the best way and now I suggest accepting the manuscript. Congratulation!